# eDNA reveals spatial differences in species composition of protected rockfishes

Stephanie A. Matthews[1]*, Olivia M. Scott[1], Meredith V. Everett[2], Megan R. Shaffer[1,2], Elizabeth Andruszkiewicz Allan[1], Andrew O. Shelton[2], Gregory D. Williams[3], Abigail Wells[2], Krista M. Nichols[2], Ryan P. Kelly[1]

**1** School of Marine and Environmental Affairs, University of Washington, Seattle, Washington, United States of America, **2** Conservation Biology Division, Northwest Fisheries Science Center, NOAA, Seattle, Washington, United States of America, **3** Pacific States Marine Fisheries Commission, Northwest Fisheries Science Center, NOAA, Seattle, Washington, United States of America

\* samatt@uw.edu

## Abstract

Rare species are difficult and time-consuming to detect, but environmental DNA (eDNA) methods can be used to increase data availability for monitoring and management. Here, we use the diverse rockfish species flock (genus *Sebastes*) to demonstrate the utility of eDNA as a tool for detecting rare and difficult to observe species in the marine environment. We describe the identification of a phylogenetically informative gene region for eDNA metabarcoding which uniquely identifies 93 of the 109 *Sebastes* species currently described. We then use this assay to differentiate rockfish communities in field samples collected from two sub-basins within Puget Sound in Washington, USA. Across three field sampling platforms, we found that sample collection location (distance from seafloor) has substantial impacts on rates of detection and on the diversity of species detected, likely reflecting the habitat preferences of the target species. This metabarcoding region provides an important tool for rockfish monitoring, both within Puget Sound and across the North Pacific. More generally, this work speaks to the usefulness of eDNA data as a tool for the conservation and management of rare and difficult-to-distinguish species.

## Introduction

Detecting rare species by any method is almost always difficult and time-consuming because of the inherent scarcity of the target species. At the same time, rare species are frequently of particular importance for monitoring efforts as they are often imperiled, or are alternatively leading-edge invasive species [1]. There is no equivalent to remote-sensing techniques for biological data, yet government bodies and managing agencies must administer living resources across vast spatial and temporal scales. Accordingly, there is substantial need for more economically efficient approaches to large-scale biological observation, in order to facilitate cost-effective monitoring and

**Data availability statement:** All data are freely available through NCBI GenBank. Raw and assembled data from genome skimming are available under BioProject PRJNA1281680. Raw MiSeq data from metabarcoding are available under BioProject PRJNA1356518.

**Funding:** This project was funded in part by the National Philanthropic Trust. The funders had no role in study design, data collection and analysis, decision to publish, or preparation of the manuscript.

**Competing interests:** The authors have declared that no competing interests exist.

management of biological resources. Environmental DNA (eDNA) offers a noninvasive, scalable method for monitoring rare and endangered species, as samples can be collected from the environment without direct observation or interaction with the target species.

The challenge of finding and distinguishing different species of rockfish is an example of the general difficulty of collecting biological information at scale. Rockfish (genus *Sebastes)* are a species flock circumglobally distributed and comprising species which are primarily deep-dwelling, bottom-associated, and sedentary [2,3]. The genus includes recently diverged sister species (<1 MYA), cryptic species pairs, and species with well-documented hybridization, all of which complicate the morphological identification of individuals [4–6]. In Washington State, Puget Sound is home to a diverse subset of *Sebastes* species, including Distinct Population Segments (DPS) of *Sebastes ruberrimus* (yelloweye rockfish) and *S. paucispinis* (bocaccio) that are listed as threatened and endangered, respectively, under the Endangered Species Act (ESA) [7]. Although the Puget Sound population of yelloweye is protected, this species is currently fished outside of Puget Sound and has historically been commercially and recreationally fished within Puget Sound [8]. Eight additional rockfishes are listed as Species of Greatest Conservation Need (SGCN) in Washington state (*S. auriculatus,* brown; *S. pinniger,* canary; *S. nebulosus,* China; *S. caurinus,* copper; *S. elongatus,* greenstriped; *S. maliger,* quillback; *S. proriger,* redstripe; *S. nigrocinctus,* tiger) [9].

Due to the difficulty of detecting bottom-associated adult individuals and accurately identifying morphologically cryptic larvae and juveniles, rockfish are a challenging model system for management by any method. At present, rockfish monitoring efforts along the west coast of the United States employ regular hook-and-line surveys, remotely operated vehicle (ROV), autonomous underwater vehicle (AUV), and other uncrewed video-based surveys, visual dive surveys, and where available, fisheries landings data [7,10]. However, eDNA monitoring offers several advantages over current techniques: morphologically cryptic sister species and morphologically unidentifiable juveniles can be assigned species identities using DNA sequences, and eDNA methods do not disturb endangered and at-risk species and do not require visual observation either by scuba divers or by video, both of which have limited spatial and temporal scales.

For *Sebastes*, molecular identification is the most reliable method for assigning species identity [11]. While nuclear panels show promise in distinguishing *Sebastes* species using tissue samples (e.g., [11]), short hypervariable markers are better suited for eDNA samples with potentially mixed compositions. eDNA techniques most often target the mitochondrial genome, as mitochondrial DNA is more abundant than nuclear DNA on a copies-per-cell basis. Due in part to greater use of mitochondrial markers, mitochondrial reference databases are also generally better populated than databases for nuclear genes. Three mitochondrial regions for identifying rockfish and discriminating between closely related species have been moderately successful. The conserved 'MiFish' region of the mitochondrial 12S gene [12] can discriminate only a handful of *Sebastes* species from one another; a somewhat more variable

mitochondrial region in the *cytochrome b* gene uniquely identifies 62 of 110 *Sebastes* species (MiSebastes; [13]). However, neither MiFish nor MiSebastes satisfactorily discriminates the 28 *Sebastes* species found in Puget Sound from one another [14]. A third recently published primer set amplifying a 206 bp region of the mitochondrial D-loop provides higher resolution for many rockfishes, but relies upon likelihood prediction to assign identities for several species with shared haplotypes [15].

Here, we describe the performance of a 356 bp region of the mitochondrial D-loop which can accurately distinguish nearly all rockfish species of concern and is still short enough for use in eDNA samples, which often contain highly fragmented DNA. We demonstrate the utility of the marker in characterizing different *Sebastes* communities in two distinct nearshore habitats, and show that, consistent with our expectation, water samples taken closer to the target-species habitats are more likely to detect the relevant species. We highlight rockfish as a useful test-case of conservation utility for these methods given the challenge of surveying rare taxa by traditional means, and the benefit of non-invasive sampling for imperiled species.

## Materials and methods

### Primer design and validation

Previous work has suggested that mitochondrial D-loop sequences differ among most species of *Sebastes* [3,16]. Early tests using the D-loop primers designed by Hyde and Vetter [3] demonstrated that they were capable of amplifying *Sebastes* eDNA from pilot samples taken in the California Bight. The Hyde and Vetter primers bind in the 3' region of the *cytochrome b* gene and approximately ~370 bp from the 5' end of the D-loop region, amplifying the tRNA-Thr and tRNA-Pro genes as well as the first ~370 bp of the D-loop. However, the amplicons generated by these primers are ~535 bp and outside the length range that returns high quality data for amplicon sequencing on the Illumina MiSeq. Using publicly available sequence data combined with preliminary sequence data from pilot eDNA samples, we designed a set of primers internal to the Hyde and Vetter primers, spanning approximately 350 bp. The newly designed forward primer is located 132 bp downstream of the Hyde and Vetter forward primer, within the tRNA-Pro gene; the reverse primer is located 83 bp upstream of the Hyde and Vetter reverse primer, within the D-loop. These primers were used to amplify a set of vouchered *Sebastes* tissues from the University of Washington Burke Museum Fish Collection, including all *Sebastes* species expected to occur in Puget Sound except *S. caurinus* (copper rockfish), as Sanger sequencing of all the tissue samples we obtained for this species had BLAST hits to *S. maliger* (quillback rockfish) [2] (Table S1 in S1 File). Amplified PCR product from each voucher specimen was Sanger sequenced at Azenta Life Sciences (Seattle, USA), and resulting sequences were used to verify species identification by BLAST pairwise alignment against the NCBI nt/nr database, as well as against the *Sebastes* database developed for this manuscript (described below). Following voucher tissue testing, the primers were tested for successful re-amplification of the pilot samples, confirming their ability to amplify eDNA from field samples.

### Database development

To determine whether all species were uniquely identified by the amplified D-loop amplicon, we developed a database of *Sebastes* species for this new primer set. Database development was accomplished in two parts: initial construction using publicly available sequence resources, followed by genome skimming [17] of vouchered *Sebastes* specimens to increase species coverage and account for any potential conspecific haplotype variation. To assemble the set of publicly available D-loop sequences, we first searched using our prospective primers using Primer–BLAST through the online NCBI portal [18]. Preliminary Primer–BLAST searches against the nt database returned matches with non-rockfish members of the Scorpaenidae and Scombridae families, but no additional matches outside of these families. As the target of interest in the present study was members of the Scorpaenidae, Scombridae hits were excluded from the reference database.

Accession numbers for each of the hits were combined into a final list (Table S2 in S1 File). We used the ReSCRIPT algorithm in QIIME2 [19] to retrieve the fasta files from the complete list. This software also retrieves the NCBI Taxonomy for each individual, and outputs it in a machine-readable format. The extract-reads algorithm in ReSCRIPT was used to trim each sequence down to only the region between the two primers.

To fill in species gaps and add unique haplotypes, we employed genome skimming on 384 vouchered *Sebastes* individuals from the northeast Pacific (Table S3 in S1 File). DNA was extracted from each individual using a DNEasy Blood & Tissue Kit (Qiagen) following manufactures instructions. Extracted DNA was quantified using the Quant-iT PicoGreen dsDNA assay kit (Invitrogen). Genome skimming libraries were created for each sample using the NEBNext Ultra II FS DNA Library Prep Kit for Illumina (NEB) following manufactures instructions and adjusting input DNA based on the quantification for each sample. The resulting libraries were individually indexed with NEBNext Multiplex Oligos for Illumina Unique Dual Index kits (NEB). The indexed libraries were cleaned with AMPure XP (Beckman Coulter*) and pooled into a final library for NovaSeq sequencing (Illumina). Sequencing was carried out at Azenta Life Sciences (Seattle, USA) on a single lane using 2 x 150 chemistry.

Sequences obtained through genome skimming were quality filtered and trimmed with cutadapt v.4.4 [20], using default quality filtering, adapter trimming on both paired ends, "G{100}" to trim sequences consisting of polyG nucleotides (an artifact of Novaseq chemistry), *--tim-n*, and *–minimum-length = 10*. Bowtie2 v2.5.0 [21] was used to align the reads to publicly available whole mitogenome sequences from *Sebastes* species (Table S3 in S1 File). Individual reference mitogenomes were converted to bowtie index files using bowtie2-build. The genome skimming sequences were assembled to these indexes using bowtie2, --local and default assembly scores. The resulting contigs were converted to fasta and bam format using samtools v.1.19 and exported along with individual bam files. Individual mitogenomes were circularized and annotated in Geneious 2024.0.5 (USA), using the bam files to confirm circularization. For species without a conspecific reference mitogenome, closely related species were chosen as references. The D-loop amplicon was extracted from each of these assemblies and added to the reference database. When gaps were found in the alignment, iterative rounds of reference alignment were carried out in Bowtie2 to attempt to fill in the gaps, using the same parameters as above, but using the contig from each iteration as the new index. In cases where sequencing coverage was low or whole mitogenome assembly could not be completed by reference-based assembly, the D-loop sequence alone was extracted and added to the reference database if possible. Raw Illumina data as well as mitogenome assemblies have been deposited in NCBI under BioProject PRJNA1281680 (for individual mitogenome accessions see Table S3 in S1 File). The complete set of sequences was dereplicated using the dereplicate algorithm in ReSCRIPT [19], and the fasta and taxonomy files were exported for use in downstream analysis. The final database included 1,380 sequences, including 1,185 publicly available D-loop sequences and 195 generated sequences from genome skimming vouchered individuals (S1 Appendix). This database was used to generate a phylogenetic tree, which was used to determine whether all species were uniquely identified at the D-loop region (S2 Appendix).

Single representative sequences from each *Sebastes* species with recorded observations in Puget Sound, together with a representative sequence from *Sebastolobus alascanus* (shortspine thornyhead)*,* were aligned using the MAFFT algorithm v7.490 [22] implemented in Geneious v2025.1.3, with the FFT-NS-i x1000 algorithm, 200PAM/k = 2 scoring, and a 1.53 and 0.123 gap open penalty and offset value, respectively. A UPGMA tree was built from the aligned sequences using the Tamura-Nei genetic distance model with 50,000 bootstrap replicates.

## Field sample collection

To assess rockfish community composition in the field, eDNA samples were collected from twelve sites within Puget Sound between March and October 2024 (Fig 1, Table S4 in S1 File). Sites were selected based on bathymetric features predicted to be rockfish habitat [23], and historical documentation by divers and anglers [24]. Water samples were collected and filtered using three approaches: scuba diver-collected samples with subsequent vacuum filtration, surface grab

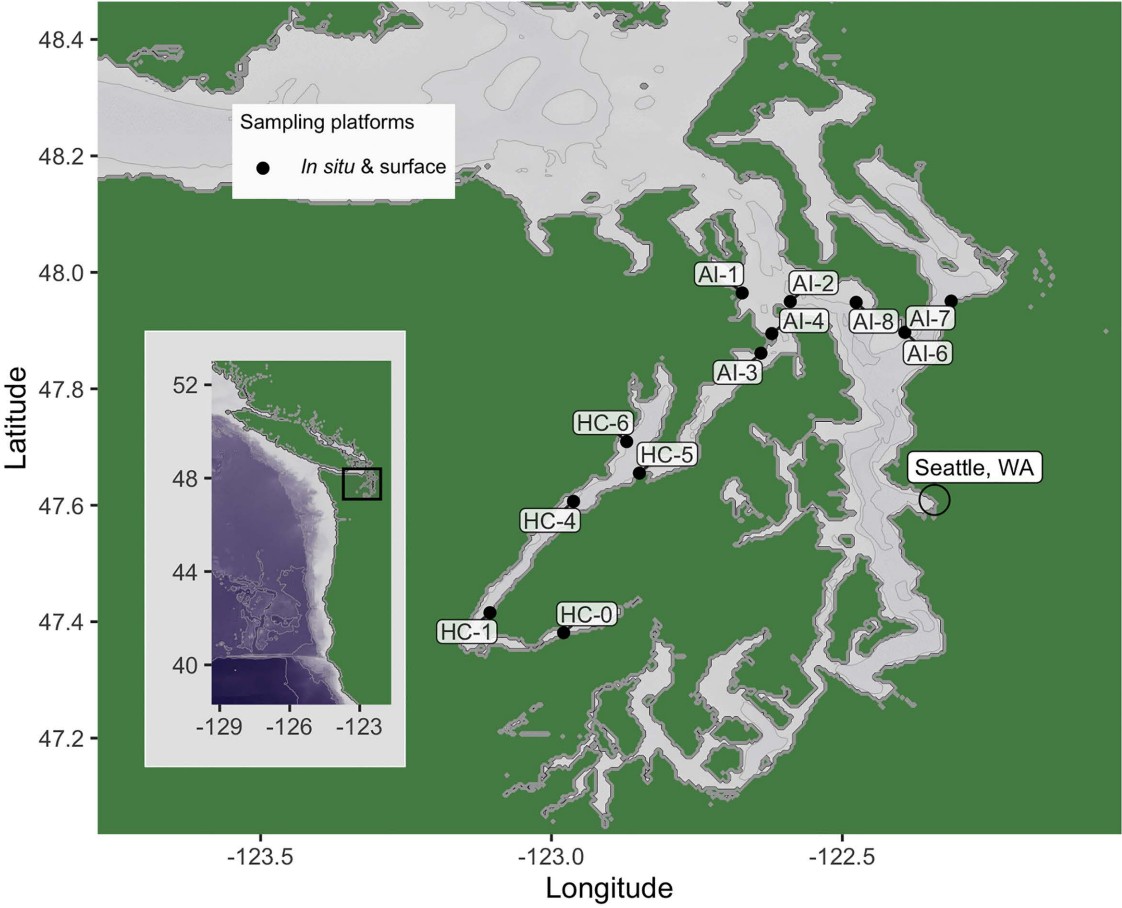

**Fig 1. Geographic locations sampled in this study.** Shapes represent the sampling platforms used at each location. Site names denote whether each site was a Hood Canal environment (HC) or Admiralty Inlet environment (AI), as shown in Fig 3. Inset shows the sampling map location relative to the west coast of North America. Seattle, WA is annotated for reference. Bathymetry and topography are sourced from the ETOPO 2022 global relief model from NOAA National Centers for Environmental Information. Bathymetry and topography data are freely available through the Creative Commons Zero 1.0 Universal Public Domain Dedication.

samples with subsequent vacuum filtration, and *in situ* filtration at depth. For diver-collected samples, two liters of water were captured at depth in a collapsible Nalgene canteen. Canteens were capped at the sampling location and transported to the surface in mesh bags. Onshore or on the boat, two liters of water was filtered onto a self-preserving 5 μm MCE filter (Sterlitech, USA) using a Citizen Scientist Sampler (Smith Root Incorporated, USA) [25]. Surface grab samples were collected in canteens at arms distance from a small boat, with samples collected as far from any boat outflow as possible. Surface samples were immediately processed as previously described for diver-collected samples. For samples that were filtered *in situ* and at depth, an Ascension eDNA sampler (Ocean Diagnostics, Canada) was fitted with pre-prepared 5μm MCE filter cartridges and deployed by hand to a depth of approximately 10 m above the seafloor at each site. At each sampling site, location (latitude and longitude) and instrument depth and distance from bottom, (detected by boat-mounted sonar and by the depth reported by the instrument's integrated conductivity, temperature, and depth (CTD) probe when available) were collected. At each sampling event (depth and location), three 2-liter samples were collected sequentially, with approximately 5–15 minutes between the start of collection times. For surface grab and *in situ* samples, filters were transported from the field location to the Center for Environmental Genomics at the University of Washington, where

filters were transferred from their housings and preserved in DNA/RNA Shield (Zymo Research). Preservation times were between 4 and 11 hours after collection, depending on the field sampling location. In total, 164 water samples were collected and filtered, with 61 samples collected by diver, 54 by surface grab, and 49 by Ascension.

No permits or ethics approval were required or obtained for this study, which involved only collection of seawater for filtration of ambient eDNA and did not interact with or disturb the study organisms.

## DNA extraction, amplification, and sequencing

DNA was extracted from filters using Quick-DNA Miniprep Plus kits* (Zymo Research). The genomic DNA from each sample was quantified by Qubit 1X dsDNA HS Assay or Quant-iT 1X dsDNA HS Assay (Invitrogen). Total genomic DNA concentrations ranged from 5.4 to 44 ng ul$^{-1}$. DNA samples were tested for inhibition using TaqMan Exogenous Internal Positive Control Reagents (Thermo Fisher), and samples with Ct values 0.5 cycles greater than the average Ct value of the no-template controls (NTCs) were considered inhibited [26]. Inhibited samples were diluted 1:10 before reanalysis, and subsequent concentration estimates adjusted accordingly. Each DNA extract was amplified with the D-loop *Sebastes* primers described above, with flanking Illumina adapters. Each PCR reaction included 4 µl of Hot-Start Platinum II Buffer, 0.16 µl of Hot-Start Platinum II *Taq*, 0.5 µl of 8mM dNTPs, 0.5 µl each of forward and reverse primers at 10 µM, 2 µl of template DNA, and 12.34 µl of water for a total PCR volume of 20 µl. PCR reactions were cycled for a 30 s denaturation at 98°C followed by 35 cycles of 10 s denaturation at 98°C, 30 s annealing at 56°C, and 30 s extension at 72°C, and a final extension step of 10 min at 72°C. Amplified PCR products were visualized on a 2% agarose gel. After bead cleaning with 1.2X ratio of AMPure XP Reagent" (Beckman Coulter) to remove residual primers and dNTPS, unique dual barcodes were added in a second PCR step. Each reaction comprised 12.5 µl HiFi HotStart ReadyMix (KAPA Biosystems, Inc), 1.25 µl DNA/RNA UD Indexes v2 (IDT for Illumina), 4–11.25 µl PCR1 product (depending on the strength of the target band in gel visualization), and water to reach 25 µl. PCR2 was cycled for a 5 m denaturation at 98°C followed by 12 cycles of 20 s denaturation at 98°C, 30 s annealing at 56°C, and 60 s extension at 72°C, and a final extension step of 10 min at 72°C. PCR products were again visualized on a 2% agarose gel. Samples were bead cleaned to remove residual primers and dNTPs with a 0.8X ratio of AMPure XP Reagent and quantified by Quant-iT 1X dsDNA HS Assay (Invitrogen). Samples were pooled equimolarly where possible, or with the maximum possible sample volume for samples with low amplicon DNA concentrations. The pooled library was then concentrated to at least 4 nM using either vacuum centrifuge or a 0.8X ratio of AMPure XP Reagent, and the fragment size of the final library was checked using a TapeStation HS D1000 ScreenTape assay (Agilent). Samples were split across three MiSeq runs, and in some cases sequenced together with unrelated samples using different primers and gene regions. Pooled libraries were sequenced in-house on an Illumina MiSeq System with V3-600 chemistry (PE 2x300), using 6–7 pM denatured and diluted library and with 20% PhiX spike-in.

## Bioinformatics and analysis

Data from each of three MiSeq runs were processed independently through a custom metabarcoding pipeline, which uses *cutadapt* for primer trimming and *dada2* implemented through R for denoising, merging of forward and reverse reads, and generation of amplicon sequence variants (ASVs) [20,27,28]. All ASVs were further filtered with *LULU* [29] to remove potential analytical artefacts such as PCR chimeras and sequencing errors. After local BLAST pairwise alignment between all ASVs, ASVs with a minimum ratio of 1, minimum match of 84, and minimum relative co-occurrence of 0.95 were collapsed into the parent ASV. The taxonomy of each ASV was assigned using the 'classifier' function from the R package *insect*, which is a recursive tree-based algorithm [30], trained on the Sebastidae database described in this manuscript with a minimum AIC threshold of 0.7. Across all samples, 12,053,667 reads were obtained, with 6,177,803 classified as non-chimeric *Sebastes* reads and 6,114,507 identified to species level. The lowest AIC value for species-level classification was 0.87, and only ASVs identified to species level were retained for further analysis. Statistical analyses used

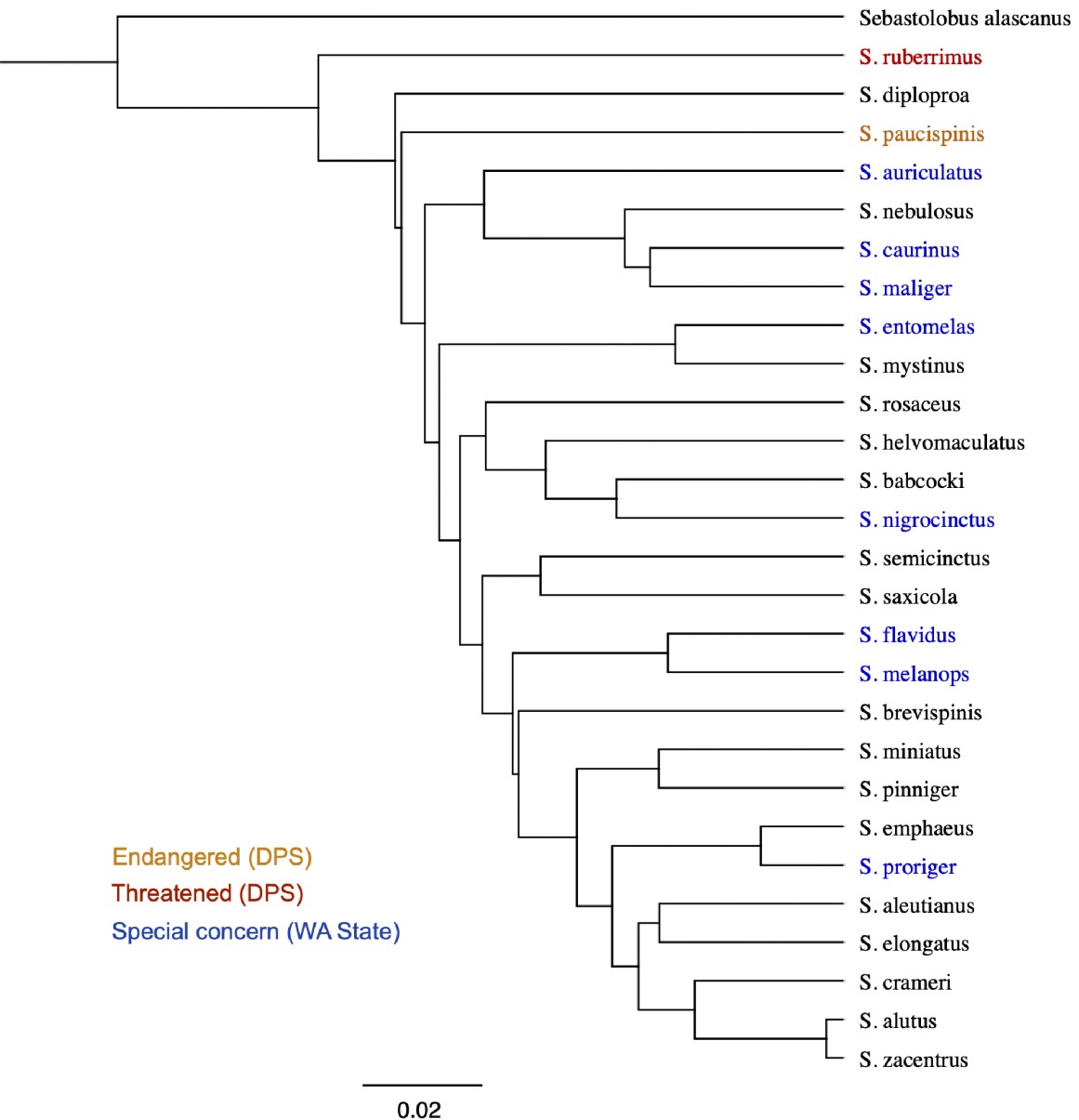

**Fig 2. Neighbor-joining tree of *Sebastes* species occurring in Puget Sound and the greater Salish Sea.** Species with Distinct Population Segments (DPS) in Puget Sound listed under the Endangered Species Act as Threatened or Endangered are highlighted in orange and red, respectively. Species of special concern in Washington State are highlighted in blue. The UPGMA tree was visualized based on MAFFT alignment. Scale bar represents genetic distance.

species level presence/absence data to avoid biases arising from amplification efficiency [31]. To test the effect of sampling depth on the number of species observed, a Poisson regression using the default natural logarithmic link function was fit using the function *stan_glm* from the package *rstanarm* [32]. Sequencing data were analyzed and visualized in R, with manipulation through the package *phyloseq* [33]. All data analysis scripts, trained *insect* classifier tree, and the final ASV table are available at https://github.com/samatthews/Rockfish_Dloop.

Modelled hydrographic data from the University of Washington LiveOcean project was obtained at hourly resolution for each field sampling location, for the calendar year 2024 [34]. One calendar year was used to characterize the seasonal variability in environment at each location, as rockfish are largely sedentary and would be expected to experience the full range of environments occurring at their location. Sites were divided into two geographic groups based on visual analysis of the 1-year variability in temperature, salinity, and oxygen values in the bottom layer of the depth-resolved model data. Topography and bathymetry data for Fig 1 were obtained from the ETOPO 2022 15 Arc-Second global relief model produced by NOAA National Centers for Environmental Information [35] using the R package *marmap* [36].

## Results

### Primer description

The newly designed primers (Rockfish-DinF: 5'-CCAAAGCCAGGATTCTTAGTTAAAC-3'; Rockfish-DinR: 5'-GCATTAAGAAATGGACTTGTTGG-3') amplify a 348 bp region within the D-loop region of the mitochondrion that successfully discriminates nearly all rockfish species, including uniquely identifying all 28 of the *Sebastes* species found in Puget Sound (Fig 2). Eight species pairs are not cleanly distinguishable at the 348 bp D-loop region (Table 1).

Across the 28 *Sebastes* species that have been recorded in Puget Sound, the described primers have 0–1 mismatches in the forward primer and 0–2 mismatches in the reverse primer for a maximum of 3 mismatches to any single species. Amplification of the target region was successful using DNA from vouchered tissue specimens of all 28 species (Burke Museum, University of Washington), indicating that all species are likely to be detected if eDNA is present in field samples. The primers also amplify members of the family Scombridae (e.g., tunas, mackerel, and bonito), of which some species are found offshore of Puget Sound but which are not commonly found within the study region.

### Rockfish detection in two habitats

Visual assessment of the near-bottom hydrographic environmental ranges at the twelve sampling sites revealed two distinct physical habitats, largely aligning with the Puget Sound basins of Hood Canal and Admiralty Inlet (Fig 3). Two stations located within Hood Canal proper (AI-3 and AI-4) but north of the Hood Canal sills were classified as Admiralty Inlet-like environments, due to greater environmental similarity with more northerly stations.

Across 12 sampling locations in two major sub-basins of Puget Sound, we detected seven unique species of rockfish (Table 2, Fig 4). Three species (*S. caurinus,* copper rockfish*; S. maliger,* quillback rockfish*;* and *S. diploproa,* splitnose rockfish) were detected in Hood Canal and seven species (*S. caurinus*, *S. maliger*, *S. auriculatus* (brown rockfish)*, S. melanops* (black rockfish)*, S. flavidus* (yellowtail rockfish) and *S. ruberrimus* (yelloweye rockfish)) were detected in Admiralty Inlet (Fig 4). *Sebastes caurinus* and *S. maliger* were the most commonly detected species (Table 2). *Sebastes auriculatus*, *S. melanops*, and *S. flavidus* were less commonly detected, while *S. diploproa* and *S. ruberrimus* were detected the fewest number of times.

Our sampling campaign incorporated three sampling platforms with partially overlapping vertical ranges. Surface and near-bottom samples were collected at all locations, as well as mid-water column samples at dive locations. We had more *Sebastes* species detections in water samples collected nearer to the seafloor (Fig 5, slope = −0.063, 95% CI [−0.084, −0.044], intercept = 0.399, 95% CI [0.107, 0.652]). We detected a maximum of one species per water sample for samples collected at the surface versus a maximum of 4 species per water sample for samples collected near the seafloor. As it was not possible to deploy all three sampling methods at the same locations or same depth ranges, we refrain from directly comparing sampling platforms.

**Table 1. Species pairs within the genus *Sebastes* which are indistinguishable using the 348 bp region amplified by Rockfish-DinF/ Rockfish-DinR. Species are listed both by their binomial and by their common name. Common names in Japanese and Korean are noted in brackets.**

| Latin names | Common names |
|---|---|
| *S. zonatus / S. vulpes* | Tanuki-mebaru [Japanese] / Gamefish |
| *S. hubbsi / S. longispinis* | Ureok [Korean] / Kôrai-yoroimebaru [Japanese]_ |
| *S. alutus / S. zacentrus* | Pacific ocean perch / Sharpchin rockfish |
| *S. ciliates / S. variabilis* | Dark dusky rockfish / Light dusky rockfish |
| *S. chrysomelas / S. carnatus* | Black-and-yellow rockfish / Gopher rockfish |
| *S. chlorostictus / S. rosenblatti* | Greenspotted rockfish / Greenblotched rockfish |
| *S. emphaeus / S. wilsoni* | Puget Sound rockfish / Pygmy rockfish |
| *S. mystinus / S. diaconus* | Blue rockfish / Deacon rockfish |

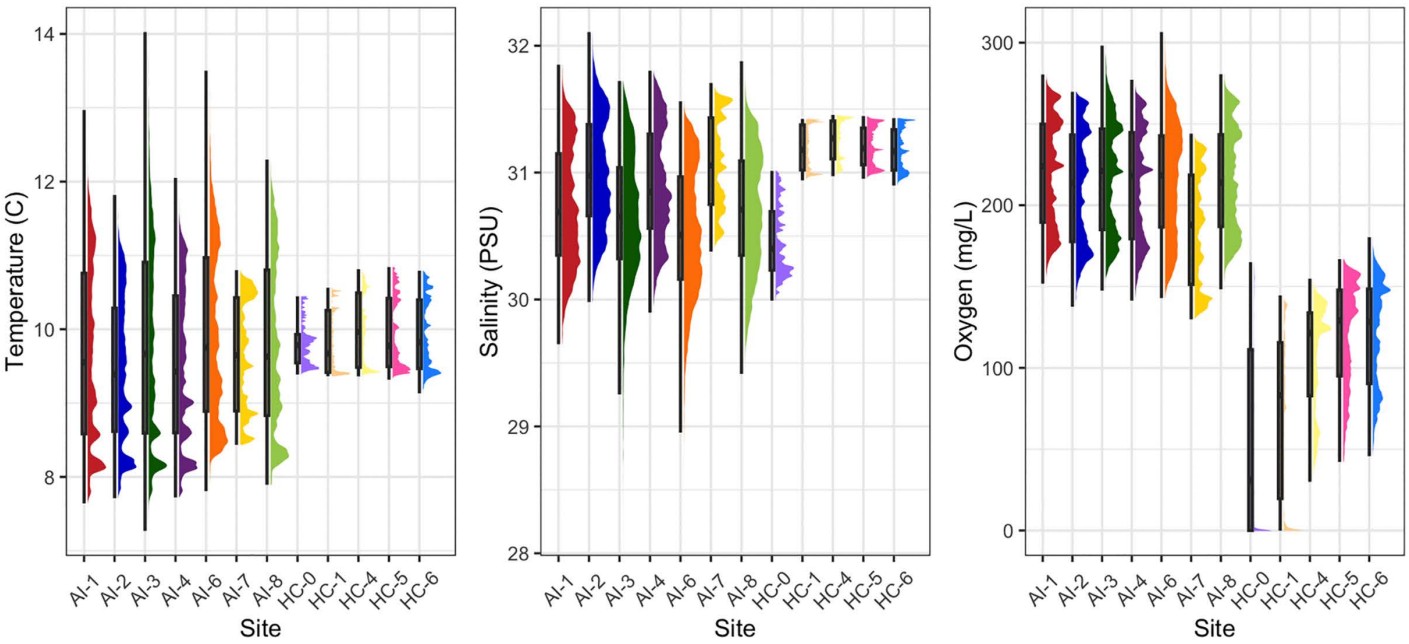

**Fig 3. Range of near-bottom environments experienced at each sampling location, for temperature (top row), salinity (middle row), and oxygen (bottom row), across the 2024 calendar year.** Environmental ranges at each sampling location were used to classify sampling sites as Hood Canal-like or Admiralty Inlet-like. Raincloud plots show the density of modeled values across the full environmental range for each station; boxplots extend from the lower quantile to the upper quantile, and whiskers indicate the total range excluding outliers. See Fig 1 for the geographic location of each site.

## Discussion

Using rockfish in Puget Sound, we demonstrate the value of eDNA as a tool for observing protected species that are difficult to detect using traditional methods. We identified a 348 bp region of the mitochondrial D-loop which can be used to identify most species in the *Sebastes* species flock. We then used eDNA metabarcoding to characterize rockfish communities in two environmentally distinct sub-basins within Puget Sound. We detected overlapping species assemblages in the two regions, including rare yelloweye rockfish in Admiralty Inlet. Rockfish diversity in any single water sample was

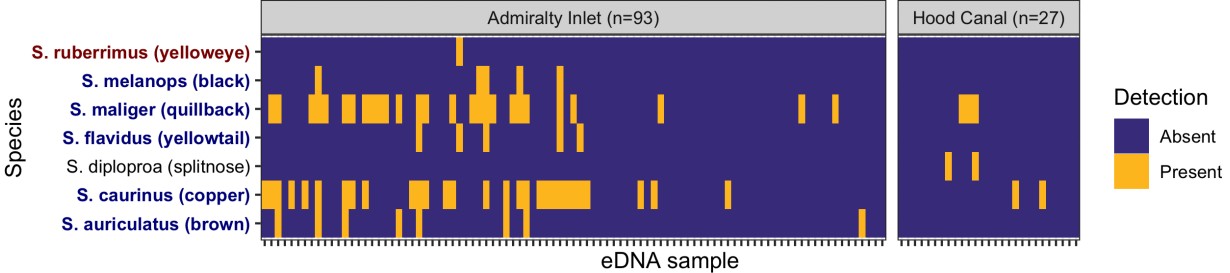

**Fig 4. Detection of each of the seven *Sebastes* species observed by metabarcoding of eDNA samples.** Ticks along the x-axis represent discrete biological samples collected in the field. Each sample indicates a unique biological replicate, including water filtered by any method, from any depth, at any station, within the two basins. Heatmap colors denote presence and absence of species in each sample. Samples collected from Admiralty Inlet and Hood Canal are shown in separate panels, following the basins denoted in Fig 2. Species names are colored by their conservation status: red = threatened (DPS); blue = special concern (WA state).

**Table 2. The number of samples and sampling sites in which each species was detected via eDNA metabarcoding, for two sub-basins of Puget Sound. Species federally listed as Threatened under the ESA are shaded in red. Species listed as SGCN in Washington State are shaded in blue.**

| Species | Hood Canal samples (#, out of 27) | Admiralty Inlet samples (#, out of 93) | Hood Canal sites (#, out of 5) | Admiralty Inlet sites (#, out of 7) |
|---|---|---|---|---|
| *S. auriculatus* (brown) | 0 | 9 | 0 | 2 |
| *S. caurinus* (copper) | 2 | 32 | 2 | 5 |
| *S. diploproa* (splitnose) | 3 | 0 | 1 | 0 |
| *S. flavidus* (yellowtail) | 0 | 6 | 0 | 1 |
| *S. maliger* (quillback) | 4 | 27 | 1 | 5 |
| *S. melanops* (black) | 0 | 7 | 0 | 1 |
| *S. ruberrimus* (yelloweye) ^ | 0 | 1 | 0 | 1 |

^ Distinct Population Segment (DPS) in Puget Sound listed as Threatened under the ESA.

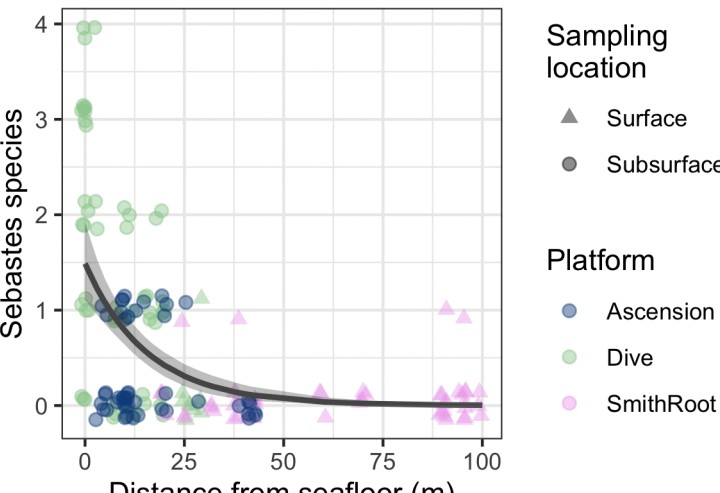

**Fig 5. The number of *Sebastes* species observed in each biological sample decreases with distance from the seafloor.** Each biological sample is shown by a partly opaque circle, colors indicate sampling platform. Points are jittered vertically for visibility; all species counts are integers. Shaded region indicates the 95% credible interval. All Smith Root samples were collected as surface grabs, across locations with different bottom depths.

negatively correlated with distance from bottom, highlighting the importance of minimizing the distance between sampling locations and putative eDNA sources. Our study shows that with appropriately designed field sampling and laboratory assays, eDNA metabarcoding can be successfully used to monitor even rare, difficult-to-observe species.

### eDNA assay performance

The region of the mitochondrial D-loop that we describe here differentiates nearly all *Sebastes* species, including uniquely identifying all *Sebastes* species that occur in Puget Sound. Of the eight species pairs not uniquely identified at the marker region, two species occur in Puget Sound. *S. diaconus* (deacon rockfish) has recently been described as a separate species from *S. mystinus* (blue rockfish) [37]. Currently only a single DNA sequence is available from vouchered specimens of *S. diaconus*, making it impossible to determine whether *S. diaconus* can reliably be differentiated from *S. mystinus*, sequences of which may include individuals of *S. diaconus* previously identified as *S. mystinus*. As *S. diaconus* extends along the West Coast of North America but has not been documented within Puget Sound, we exclude it from the current analysis. *S. emphaeus* (Puget Sound rockfish) and *S. wilsoni* (pygmy rockfish) are similarly indistinguishable based on current sequences. However, *S. wilsoni* is not known to occur in Puget Sound and so sequences originating from Puget Sound matching either of these species can be assumed to originate from *S. emphaeus* [31,38].

Previously published metabarcoding markers for fishes, and for rockfishes specifically, have not satisfactorily differentiated the rockfish species found in Puget Sound. MiFish [12] uniquely identifies only five of the twenty-eight rockfishes with distributions including Puget Sound, and Misebastes [13] only seventeen. The recently described 206 bp D-loop marker [15], which falls within our 348 bp region, performs significantly better, but fails to differentiate six of the twenty-eight rockfish species found in Puget Sound, when assessed within our D-loop database. The metabarcoding assay described here is 348 bp and located in the highly variable, non-coding D-loop, providing greater resolution between co-occurring *Sebastes* species and fully differentiating all rockfishes with known occurrences within Puget Sound. Because the assay detects mitochondrial DNA which is inherited only through the maternal line, it does not have the capacity to differentiate species hybrids. In the future, development of nuclear eDNA assays may allow for more targeted investigation of hybridization [39].

### Rockfish in Puget Sound

We detected seven rockfish species across two sub-basins of Puget Sound. Rockfishes in Puget Sound comprise 28 species in total, five of which are relatively to very common (Table S6 in S1 File) [24,40–42]. We detected all five of these more common rockfish in Puget Sound (copper, quillback, brown, black, yellowtail). All five species have a short pelagic stage either as young-of-the-year or juveniles. Copper, quillback, and brown rockfish are largely benthopelagic as adults, while black and yellowtail exhibit more pelagic schooling behavior. Our samples were collected in March, April, and October in Admiralty Inlet, and October in Hood Canal (Fig 1). All five common species can be observed as larvae or young-of-the year in Puget Sound during these months, with brown, black, and copper typically settling during the summer gap between sampling events. All five of the more common species were either mostly or entirely detected near bottom, apart from detection of copper rockfish in two near-surface samples in March, and copper and brown each in one near-surface sample in October. Given the lack of repeated sampling at any location, we cannot determine whether the near-surface detections stem from pelagic life history stages or from adult rockfish, either through vertical eDNA transport or fish forays into the pelagic habitat. More extensive field sampling across multiple seasons, which is more feasible for eDNA surveys than for traditional diver or hook-and-line surveys, would help distinguish between seasonal and spatial patterns in rockfish distributions.

In addition to the more common species, we also detected two moderately rare rockfish species: *S. diploproa* (splitnose rockfish) and *S. ruberrimus* (yelloweye rockfish). In our data, splitnose was observed at one location in Hood Canal, both

in surface and near-bottom samples. Within Puget Sound, splitnose has been most commonly observed in the juvenile stage, and within Hood Canal it may be nearly as common as copper, black, and yellowtail (D. Lowry, pers. comm). Previous reports of *S. diploproa* in the Hood Canal are of juveniles associated with floating kelp rafts, corroborating our surface detections [24,43]. Our detection of yelloweye rockfish was restricted to a single location in Admiralty Inlet, in a near-bottom sample. Yelloweye rockfish have historically been among the more common species in Puget Sound [24], but the population is depleted due to overfishing [44] and the species is now relatively rare in the region. Yelloweye are highly sedentary as adults, and our detection of this species particularly exemplifies the power of eDNA for detecting rare species of concern.

In our mixed template eDNA samples, we detected a maximum of four species in any single water sample. As we do not currently have amplification data from a mock community for calculation of amplification efficiency [31,38], we cannot interpret the proportions of different species within a sample and rather present our data as binary presence/absence observations only. However, the D-loop primers described here have up to three mismatches with recorded sequences from rockfish found in Puget Sound. The presence of mismatches can negatively impact amplification efficiency, and in extreme cases can even result in non-detection as amplicons with more efficient amplification begin to dominate the metabarcoding reads [31]. For our primer set, mismatches are unlikely to cause such extreme effects as complete drop-out, as the impacts of mismatches dominate only at higher numbers of mismatches (> 6) [38]. The total rockfish species richness we detected is certainly an underestimation of rockfish species richness in Puget Sound, as there are 21 additional species of rockfish with documented observations in Puget Sound which we did not detect in our field sampling. Many of these species are observed only rarely (Table S6 in S1 File), or are associated with habitats we did not sample.

## Importance of appropriate field sampling

For rare, sedentary species of interest such as rockfish, our data reveal that sample collection location can have substantial impacts on probability of detection. We used a combination of field sampling methods requiring a range of effort, including relatively easy surface water grabs, more time consuming at-depth filtration of water, and diver-collected water samples from near known rockfish habitat. Surface water grabs are commonly used for eDNA analysis, due to ease of collection. For many species, surface grabs have satisfactory detection rates. However, here our target of interest, rockfishes, are bottom-associated for the most part [2]. Accordingly, we found higher detection rates for near-bottom samples, and the number of rockfish species detected increased with proximity to the bottom. We observed the highest richness in diver-collected samples, which were both horizontally and vertically located within 1–2 meters of known rockfish habitat. Ascension-collected samples were similarly targeted above sites with recorded rockfish hook-and-line catches, but we were unable to visually verify that the instrument was deployed directly above the rockfish habitat. Additionally, we targeted the Ascension instrument deployment to approximately 5 m above the seafloor, to minimize the risk of contacting the sea floor or rocky outcroppings in the often high-relief sampling locations. Variability in eDNA detection rates across water collection methods are primarily associated with differences in the volume of water filtered, filter mesh size, preservation method, and proximity to eDNA source [45]. We cannot rule out differential detection rates among our water collection methods, as there is insufficient overlap in the depths and geographic locations sampled by each method. However, we expect the observed differences among platforms in our study are driven by variability in proximity to eDNA sources, as the mesh size, volume filtered, and preservation method were consistent across methods.

eDNA has been successfully used to map rockfish in the past, with results dependent on the resolution of the metabarcoding marker used. Sequences at the MiSebastes marker (184 bp of *cytochrome b*) showed species distributions that are comparable to diver surveys, while reflecting overall greater diversity relative to visual identifications [14]. Similarly, eDNA samples collected alongside net hauls detected comparable species compositions, with the absence of species inhabiting only depths deeper than the depth of eDNA collections [15].

The sparsity of our rockfish detections, both at surface and depth, are likely influenced by a combination of biological and physical factors. Previous work has shown that across horizontal transects, community compositions derived from eDNA metabarcoding are distinct on scales of tens of meters [46]. Our sampling locations ranged from 20–100 m in depth, in a location heavily influenced by fresh-water input and with strong vertical density gradients which can reduce vertical mixing [34]. Many rockfish species are bottom-associated for most of their lifespan and exhibit largely sedentary behavior which may reduce DNA shedding relative to more active pelagic fishes [47]. Together, these factors likely explain the few surface detections in our data. In environments with stronger sub-mesoscale mixing (e.g., unprotected coastal habitats), depth of sampling may have a smaller effect on detection rates.

## Conclusion

eDNA has been identified as a promising method for rapid, scalable surveys and monitoring of rare and hard-to-detect species. However, eDNA is seldom incorporated as a routine monitoring tool as of yet. The metabarcoding and eDNA assay described here provides the best species resolution to date for rockfish, uniquely identifying all *Sebastes* species found in Puget Sound, and approximately 85% of all presently described rockfish species. Our application of the assay in Puget Sound provided valuable and previously unavailable data on rockfish occurrences in Puget Sound, even for a relatively small field sampling campaign. Taken together with emerging quantitative techniques (e.g., qPCR, ddPCR), eDNA metabarcoding can quantitatively assess rockfish distributions and community composition [48]. In addition to contributing a valuable tool for rockfish surveys, our work demonstrates the usefulness of eDNA as a tool for researchers and managers generally. With appropriate assay design and with eDNA sample collection designed for the habitat and behavior of the target species, eDNA metabarcoding provides valuable data that otherwise would be unavailable or prohibitively expensive.

## Supporting information

**S1 File. Supporting tables.** Tables S1–S6.
(XLSX)

**S1 Appendix. D-loop database in aligned FASTA format.**
(ZIP)

**S2 Appendix. Neighbor-joining tree of D-loop database.**
(ZIP)

## Author contributions

**Conceptualization:** Stephanie A. Matthews, Meredith V. Everett, Elizabeth Andruszkiewicz Allan, Krista M. Nichols, Ryan P. Kelly.

**Data curation:** Stephanie A. Matthews, Meredith V. Everett.

**Formal analysis:** Stephanie A. Matthews, Meredith V. Everett.

**Funding acquisition:** Elizabeth Andruszkiewicz Allan, Krista M. Nichols, Ryan P. Kelly.

**Investigation:** Stephanie A. Matthews, Olivia M. Scott, Meredith V. Everett, Megan R. Shaffer, Andrew O. Shelton, Gregory D. Williams, Abigail Wells.

**Methodology:** Stephanie A. Matthews, Olivia M. Scott, Meredith V. Everett, Megan R. Shaffer, Elizabeth Andruszkiewicz Allan, Andrew O. Shelton, Gregory D. Williams.

**Project administration:** Stephanie A. Matthews, Meredith V. Everett, Elizabeth Andruszkiewicz Allan, Krista M. Nichols, Ryan P. Kelly.

**Resources:** Krista M. Nichols.

**Validation:** Stephanie A. Matthews.

**Visualization:** Stephanie A. Matthews.

**Writing – original draft:** Stephanie A. Matthews.

**Writing – review & editing:** Stephanie A. Matthews, Olivia M. Scott, Megan R. Shaffer, Elizabeth Andruszkiewicz Allan, Andrew O. Shelton, Gregory D. Williams, Abigail Wells, Krista M. Nichols, Ryan P. Kelly.

## Acknowledgments

We thank Jameal Samhouri for his ongoing programmatic support. Selection of sampling locations was informed by data provided by Kathryn C. Meyer and Robert E. Pacunski, WDFW. Thanks to Piper Schwenke and Jason Dobry for initial primer development and testing. Katherine Maslenikov at the University of Washington Burke Museum Fish Collection and the NOAA Hook & Line Survey provided valuable tissue samples for primer development and *in vitro* assay testing. Kelly Andrews, Dayv Lowry, and Bob Pacunski helped qualitatively rank rockfish species abundances in Puget Sound. Two anonymous reviewers provided helpful feedback which improved the legibility of the manuscript.

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
