## [Decision Letter · Decision Letter 0]

5 Aug 2025

eDNA reveals spatial differences in species composition of protected rockfish

PLOS ONE

Dear Dr. Matthews,

Thank you for submitting your manuscript to PLOS ONE. After careful consideration, we feel that it has merit but does not fully meet PLOS ONE’s publication criteria as it currently stands. Therefore, we invite you to submit a revised version of the manuscript that addresses the points raised during the review process.

We look forward to receiving your revised manuscript.

Kind regards,

Masami Fujiwara, PhD

Academic Editor

PLOS ONE

Journal Requirements:

“This project was funded in part by the National Philanthropic Trust.”

6. We note that Figure 2 in your submission contain map/satellite images which may be copyrighted. All PLOS content is published under the Creative Commons Attribution License (CC BY 4.0), which means that the manuscript, images, and Supporting Information files will be freely available online, and any third party is permitted to access, download, copy, distribute, and use these materials in any way, even commercially, with proper attribution. For these reasons, we cannot publish previously copyrighted maps or satellite images created using proprietary data, such as Google software (Google Maps, Street View, and Earth). For more information, see our copyright guidelines: http://journals.plos.org/plosone/s/licenses-and-copyright.

 1. You may seek permission from the original copyright holder of Figure 2 to publish the content specifically under the CC BY 4.0 license. 

Additional Editor Comments:

First, I would like to apologize for the delay in the review process. Initially, two reviewers agreed to evaluate the manuscript, but only one submitted their comments. We then had two additional reviewers, and again received comments from just one. Fortunately, we now have feedback from two reviewers who are well-versed in eDNA methods and their application to marine fish detection. 

Both reviewers are generally positive about the manuscript and commend your efforts in tackling a technically challenging topic. However, they raised several important concerns that will need to be addressed. I am recommending a **Major Revision**  to ensure the revised manuscript fully satisfies the reviewers' expectations.

**Key Points to Address:**

**Hybrid Detection and Mitochondrial Limitations**One reviewer noted that while hybridization is briefly mentioned, the manuscript does not sufficiently discuss its implications. Given the use of mitochondrial markers, which only reflect maternal lineages, I suggest including a paragraph in the Discussion that addresses:The challenges of detecting hybrids with eDNA.Potential biases in species richness estimates.Relevant literature on this topic.**Bioinformatics Transparency**There is concern about the lack of detail regarding bioinformatics methods. This is particularly important for a methodological journal like *PLOS ONE* . Please ensure all tools, parameters (e.g., trimming thresholds, software versions), and workflows are fully and clearly described.**Mock Community Assay**One reviewer highlighted the absence of a mock community experiment as a limitation, especially regarding primer efficiency and species detectability. As this aspect falls outside my expertise, I will defer to the reviewers to evaluate your response to this point in the revised submission.

 In addition, there are several constructive suggestions related to figures, minor text edits, and data accessibility. Please incorporate these changes where appropriate. If you choose not to implement a particular suggestion, provide a brief rationale in your response to reviewers. The reviewers have contributed thoughtful and detailed feedback, and I am confident they would appreciate careful consideration of their comments.

The goal is to strengthen the manuscript so that it makes a greater impact within the scientific community.

Reviewers' comments:

Reviewer's Responses to Questions

**Comments to the Author**

1. Is the manuscript technically sound, and do the data support the conclusions?

Reviewer #1: Yes

Reviewer #2: Yes

2. Has the statistical analysis been performed appropriately and rigorously?

Reviewer #1: Yes

Reviewer #2: N/A

3. Have the authors made all data underlying the findings in their manuscript fully available?

Reviewer #1: No

Reviewer #2: Yes

4. Is the manuscript presented in an intelligible fashion and written in standard English?

Reviewer #1: Yes

Reviewer #2: Yes

Reviewer #1: This study demonstrates the power of eDNA, especially for a genus of fishes of commercial and ecological importance. It clearly contributes to our understanding of biodiversity in the Puget Sound region and has important implications for conservation. The findings show that eDNA samples collected from benthic regions yield higher species counts compared to surface samples and more accurately reflect where species were detected.

I have one major reservation about this study. Although the authors acknowledge that hybridization occurs within this genus, they do not address the limitations of detecting hybrids through eDNA sampling. This is particularly important when using mitochondrial loci, which only provide information on the maternal lineage. While the authors cite Schwenke et al. (2018) on page 3—where it is suggested that hybridization may be occurring in Puget Sound—this point is not further discussed in the manuscript.

I strongly recommend adding a paragraph in the discussion and conclusion sections to address the caveats of using eDNA, including the potential for underestimating species richness. Additionally, please include citations that discuss the limitations of eDNA and how integrating nuclear DNA data can help resolve hybridization issues.

Another helpful addition would be a flowchart illustrating the database development process. This could be included as supplemental information to clarify how primers were generated and applied.

Below are minor comments to further improve the manuscript.

Line 52: china should be changed to China. Since the species name is the name of a country as well.

Line 129: What are the version numbers for all bioinformatics programs (i.e. cutadapt, bowtie, phyloseq, etc)?

Line 231: There was not much information about data analysis scripts completed on the GitHub site.

Line 246: For consistency, remove spaces from S. hubbsi / S. longispinis to match other species pairs.

Reviewer #2: The work by Matthews and collaborators reports the development of a metabarcoding assay to discriminate most of the rockfish species (genus Sebastes).

This is a very speciose genus, with many of its species inhabitting the same waters. Correctly detecting the presence of a species is very difficult, and doing so, regardless of the methods, requires a lot of effort.

The authors target the Dloop zone, and design a new set of primers to amplify a ~350 bp fragment able to discriminate almost all Sebastes species, and all of those expected in the Puget Sound (PS) and Hood Canal (HC). To do so, they increase the coverage of the Sebastes spp in public databases, by sequencing their genomes and assembling mitogenoms with genome skimming.

The authors design the primers and with the amplicons generate a classification tree using the R package insect. Then they test the assay with eDNA samples from HC and PS, detecting 7 rockfish species.

What are the standards needed to clear when designing a new assay? The authors use a set of them:

-Primer binding: One template tests for the species expected in HC and PS. Reporting positive results, unclear if successful amplicons were sequenced to confirm identity.

-Discrimination in silico

-Working with real samples

The one thing I am missing is a mock community assay, in which you test whether the relative performance of the primers in different taxa might deem some species undetactable, given that after 35 PCR cycles the relative proportion of amplicons from species with lower afinity might be too small to be detected.

This could help explain the low proportion of rockfish species detected (7 out of the 28 present, in 12 sites) - but maybe they are rare species and thus harder to detect.

Another possibility is that the bioinformatics pipeline, particularly at the LULU stage, is merging true ASVs into one OTU.

At the time of reviewing we don't have the ASV data available, nor the classification tree, so I can't

Besides that, the article is well reasoned and written, with a few inline comments:

line 35: "state and federal governments" These are government bodies in the United States - rephrase to make it applicable to other countries (eg goverment bodies, managing agencies)

line 36-37: The sentence sounds incomplete - please elaborate further.

line 43 : Is species flock an accepted term?

Sentence from line 70 to 73. They way it is written seems to point at the large number of copies and database coverage as the reason these hypervariable regions are mitochondrial. I think the reason is that the mitochondrial DNA evolves at a higher rate than nuclear DNA(https://doi.org/10.1093/molbev/msx197), and the control region does it at an even higher rate. Plus the heredability of the mitochondrial genome stops recombination and substitutions are the likely pattern of evolution.

Line 82: probably another good reason is the length limitiation of the high quality NGS platforms.

line 126: Typo in sequencing

line 128: You need to estate the trimming paremeters - particularly with the Qscore binning in NovaSeq data

line 129: You used cutadapt but no adapter trimming was made, correct?

line 160 : ...in situ and at depth, an Ascension eDNA sampler...

FIGURE 1: The figure is indeed a map, but a very hard one to read. Can you re do the figure with easier color selections (one color for all terrain above water, maybe a window to the location of the sampling area with respect to North America's West Coast, remove "Downtown") so it is easier to read?

line 186: Spell out No template Control

line 190: We know there are 8 species pairs sharing reference sequences, but we don't know the minimum number of differences between species. Wouldn't it be better to find a suitable HiFi Taq for the whole process?

line 218: Lulu's documentation as per the reference provided needs a match table between all detected OTUs - did you do that or does it work differently in a newer package?. I would prefer the ASV table and sequences to be shared, after and before Lulu.

line 221: Need to provide the AIC thresold you allowed insect to work with. I assume most of the assignments had a score of NA (as the query sequence was already present in the database), but if there are many low values, maybe you are allowing species-level asignments when the algorithm is not too confident. report the cutoff value to get a sense of this. Computing the classification tree requires a lot of time and computer power. Are you sharing the tree as an .rds file?

FIGURE 2: NJ tree is ok to reflect sequence similiarity, as it groups similar sequences first, and the phylogenetic relationship amongst all Sebastes species are beyond the scope of the paper. Despite that, both the MAFFT aligment and NJ tree specifications deserve to be in the method section.

line 330: I agree that the region differentiaties the deposited sequences, but only 7 species have been amplified and detected. One very possible explanation is the local absence of the species that did not show up, but another one could be that your primers do not bind very well. I know in table S1 you show the species by species tests, but what about if species are co-ocurring in a sample? Relative affinity to the primers could make the signal from the less favored species dissapear. You show that there are only between 0 and 3 mismatches between primers and deposited sequences, so maybe these differences do not account for much. I do not have enough evidence to go one way or the other, and only a mock community test can solve this. Same for the conclusion in lines 413-415.

line 351: 5 of which are relatively to very common. Spell out Five, also in line 352

Regarding your data sharing and availability, please clarify that you'll be sharing:

-raw sequence data from genome skimming

-raw sequence data from metabarcoding MiSeq runs

-Insect classification tree used to identify the sequences generated

-ASV tables and sequences before and after Lulu

**Do you want your identity to be public for this peer review?** For information about this choice, including consent withdrawal, please see our Privacy Policy

Reviewer #1: No

Reviewer #2: No

---

## [Author Response · Author response to Decision Letter 1]

30 Sep 2025

The response to reviewers has been uploaded as a .docx file.

---

## [Decision Letter · Decision Letter 1]

27 Oct 2025

Dear Dr. Matthews,

Thank you for submitting your manuscript to PLOS ONE. After careful consideration, we feel that it has merit but does not fully meet PLOS ONE’s publication criteria as it currently stands. Therefore, we invite you to submit a revised version of the manuscript that addresses the points raised during the review process.

We look forward to receiving your revised manuscript.

Kind regards,

Masami Fujiwara, PhD

Academic Editor

PLOS ONE

Journal Requirements:

Reviewers' comments:

Reviewer's Responses to Questions

**Comments to the Author**

Reviewer #1: All comments have been addressed

2. Is the manuscript technically sound, and do the data support the conclusions?

Reviewer #1: Yes

3. Has the statistical analysis been performed appropriately and rigorously?

Reviewer #1: Yes

4. Have the authors made all data underlying the findings in their manuscript fully available?

Reviewer #1: No

5. Is the manuscript presented in an intelligible fashion and written in standard English?

Reviewer #1: Yes

Reviewer #1: This study by Matthews et al. provides a valuable resource for monitoring the presence or absence of multiple rockfish species using eDNA. All of my major concerns and suggestions were satisfactorily addressed in the revised version of the manuscript, particularly those regarding the limitations of mitochondrial markers for eDNA applications. However, I believe several minor edits should still be addressed before publication. My specific comments and suggestions are listed below.

General comments

The resolution of the figures is low; please provide higher-resolution versions for publication quality.

There is inconsistency in citation formatting—some references are numbered, while others use author–year style. Please ensure the citation format is consistent throughout.

Line-by-line comments:

Line 18: Remove italics from the closing parenthesis “)”.

Line 51: Correct the spelling of bocaccio.

Line 62: Spell out ROV (Remotely Operated Vehicle) and AUV (Autonomous Underwater Vehicle), as not all readers may be familiar with these abbreviations.

Line 101: The D-loop primers from Hyde and Vetter (2007) are mentioned, but their positions relative to the primers developed in this study are unclear. Please clarify this comparison.

Line 105: Provide the name of the exceptional rockfish species mentioned here.

Line 169: Use the full genus name (Sebastolobus alascanus) since this is the first mention of the species.

Line 190: Spell out CTD (Conductivity, Temperature, Depth).

Line 284: Sebastes should be italicized.

Line 296: Include examples of fishes in the family Scombridae (e.g., tunas, mackerel, and bonito).

Lines 371–372: Ensure consistency in listing species names and their common names. When a species is first mentioned, include its common name.

Line 396: Consider referring to Figure 1 here, as Admiralty Inlet and Hood Canal are discussed in this section.

Line 458: Insert a space between “184” and “bp.”

Figure 2 legend: Include a note indicating that the scale bar represents genetic distance (e.g., “Scale bar = 0.02 genetic distance”).

**Do you want your identity to be public for this peer review?** For information about this choice, including consent withdrawal, please see our Privacy Policy

Reviewer #1: No

---

## [Author Response · Author response to Decision Letter 2]

7 Nov 2025

The response to reviewers has been uploaded as a separate document.

---

## [Editor Report · Decision Letter 2]

12 Nov 2025

eDNA reveals spatial differences in species composition of protected rockfishes

PONE-D-25-30427R2

Dear Dr. Matthews,

We’re pleased to inform you that your manuscript has been judged scientifically suitable for publication and will be formally accepted for publication once it meets all outstanding technical requirements.

Kind regards,

Masami Fujiwara, PhD

Academic Editor

PLOS ONE
---

## [Editor Report · Acceptance letter]

PONE-D-25-30427R2

PLOS ONE

Dear Dr. Matthews,

I'm pleased to inform you that your manuscript has been deemed suitable for publication in PLOS ONE. Congratulations! Your manuscript is now being handed over to our production team.

Kind regards,

on behalf of

Dr. Masami Fujiwara

Academic Editor

PLOS ONE